# Polyacrylonitrile-*b*-Polystyrene Block Copolymer-Derived Hierarchical Porous Carbon Materials for Supercapacitor

**DOI:** 10.3390/polym14235109

**Published:** 2022-11-24

**Authors:** Ainhoa Álvarez-Gómez, Jiayin Yuan, Juan P. Fernández-Blázquez, Verónica San-Miguel, María B. Serrano

**Affiliations:** 1Department of Materials Science and Engineering and Chemical Engineering (IAAB), University of Carlos III of Madrid, Av. Universidad, 30, 28911 Leganés, Spain; 2Department of Materials and Environmental Chemistry, Stockholm University, 10691 Stockholm, Sweden; 3IMDEA Materials Institute, C/Eric Kandel 2, 28906 Getafe, Spain

**Keywords:** block copolymer template, porous carbon fibers, hierarchical pores, supercapacitor

## Abstract

The use of block copolymers as a sacrificial template has been demonstrated to be a powerful method for obtaining porous carbons as electrode materials in energy storage devices. In this work, a block copolymer of polystyrene and polyacrylonitrile (PS−*b*−PAN) has been used as a precursor to produce fibers by electrospinning and powdered carbons, showing high carbon yield (~50%) due to a low sacrificial block content (*f*_PS_ ≈ 0.16). Both materials have been compared structurally (in addition to comparing their electrochemical behavior). The porous carbon fibers showed superior pore formation capability and exhibited a hierarchical porous structure, with small and large mesopores and a relatively high surface area (~492 m^2^/g) with a considerable quantity of O/N surface content, which translates into outstanding electrochemical performance with excellent cycle stability (close to 100% capacitance retention after 10,000 cycles) and high capacitance value (254 F/g measured at 1 A/g).

## 1. Introduction

Supercapacitors (SCs) are electrochemical devices that store energy by intercalating charges at the electrode−electrolyte interface. SCs development has increased in the last decade due to the higher capacitance and lower voltage limits compared to conventional capacitors [1]. From an energy storage point of view, SCs bridge the gap between electrolytic capacitors and batteries [2]. The main advantages of these devices over other systems include fast charge−discharge (on the level of seconds) without losing efficiency or degrading their internal structures, high power densities (>1 W/g), low heat generation, and long lifetime (>500,000 cycles) due to the storage mechanism, which does not involve irreversible reactions [3]. Nevertheless, supercapacitors present considerably lower energy density than conventional batteries, which limits their use in applications, such as electric vehicles, that demand high energy and power density.

Owing to their combination of outstanding physical and chemical properties such as high electrical conductivity, high surface area, good corrosion resistance, and thermal stability, activated carbons have been the most widely used and commercially available electrode materials for electrochemical double−layer capacitors (EDLCs) [4,5]. In particular, high conductivity and high surface area are considered key features in high−performance electrode materials [6]. However, active carbons present some limitations that must be overcome regarding their applications in supercapacitors. Although high surface areas (>1000 m^2^/g) are achieved, high capacitance values are not necessarily obtained since not all of the micropores are accessible to the electrolyte ions [7]. The reason for this is that these materials do not usually show precise control over the porous structure due to the difficulty of optimizing the activation processes [8]. Pore size and pore size distributions have been found to strongly influence electrochemical performance (i.e., the accessibility of ionic species, capacitance, energy density, and power density) [9]. Therefore, the development of advanced carbon electrode materials is largely focused on designing and obtaining precise carbon nanostructures [10,11]. Hierarchical porosity comprising small and large mesopores (2–50 nm), which facilitate electrical double-layer formation (as well as macropores) has been shown to improve ion−accessibility and ion diffusion, allowing easy access to the micropores (<2 nm) [12].

Due to their high versatility and various synthetic routes that allow excellent control over molecular weight [13,14], block copolymers (BCPs) are ideal precursors for obtaining ordered nanostructured materials, as the controlled morphologies generated by the microphase−separation and self−assembly can be retained after pyrolysis [15,16]. The use of BCPs as templates (soft templating) has proven to be an effective method for obtaining porous carbon materials suitable for energy storage applications [17,18]. BCPs used as carbon precursors typically consist of a high carbon−yielding polymer as carbon matrix, usually polyacrylonitrile (PAN), and a thermally degradable sacrificial block. These block copolymers can be easily converted into porous carbon materials derived through subsequent thermal treatments [19]. A critical first step to ensure a high carbon yield is to stabilize PAN at temperatures around 300 °C in an oxidizing atmosphere, followed by carbonization under nitrogen. These thermal treatments allow one to retain the phase-separation morphologies and form the final porous carbon structures. Some of the most commonly used sacrificial blocks include poly(butyl acrylate) (PBA) [20], polystyrene (PS) [21], poly(ethylene oxide) [22], poly(acrylic acid) (PAA) [23], and poly(methyl methacrylate) (PMMA) [24].

Among different porous carbon structures, such as monoliths [25] or membranes [26], porous carbon fibers (PCFs) stand out for offering a combination of low density, freestanding nature, flexibility, good chemical stability, and excellent thermal and electrical conductivity [27,28]. PCFs have been successfully designed from diverse materials displaying high specific surface area (SSA) with excellent electrochemical performance [29]. Compared to powder carbons, fibers present two main advantages [30]: (i) the surface area accessible to the electrolyte is increased; and (ii) one−dimensional fibers provide continuous electron conduction pathways with small electrical resistance. In addition, to boost their electrochemical applications, ideal PCFs should possess hierarchical porous structures with mesopores and micropores interconnected. Although different polymer blends have been explored as template materials for obtaining PCFs as supercapacitor electrodes [31,32,33], BCPs have been recently proven to be a powerful new precursor for the fabrication of PCFs for electrochemical performance. G. Liu et al. reported one of the highest capacitance values found in the literature [34] using PMMA−*b*−PAN copolymer-based porous carbon fibers for supercapacitors. Therefore, exploring different copolymers to find optimal nanostructures and, consequently, improve their electrochemical performance elicits noteworthy attention. This article focuses on the structural and morphological comparative study of two different electrode materials for supercapacitors, constituted by PCFs based on PAN−*b*−PS copolymer with a hierarchical porous structure and the carbon powder as bulk material without any processing. (Figure 1). 

## 2. Materials and Methods

### 2.1. Materials

Acrylonitrile (AN, ≥99%) and styrene (≥99%) were purchased from Sigma-Aldrich (Saint Louis, MO, USA) and purified by passing through a basic alumina column in order to remove the inhibitor before use. (2,2′−azobis(2−methylpropionitrile) (AIBN) (≥98%; Cymit), 2−cyano−2−propyl dodecyl trithiocarbonate (CPDT; ≥98%; Sigma-Aldrich), anhydrous *N,N*−dimethylformamide (DMF; ≥99.8%; Sigma-Aldrich), tetrahydrofuran (THF; ≥99%; Sigma-Aldrich), and methanol (MeOH; ≥99.8%; Sigma-Aldrich) were used as received without further purification.

### 2.2. Synthesis of PS−CPDT MacroCTA

PS−*b*−PAN copolymer was synthesized by reversible addition-fragmentation chain transfer polymerization (RAFT) using AIBN as initiator and CPDT as chain transfer agent (CTA), via two steps (Appendix A). First, a macroCTA (PS−CPDT) was synthesized. Based on previously reported [35], a typical procedure was carried out as follows: styrene (2 mL, 17 mmol) and CPDT (30 µL, 0.085 mmol) were added to a Schlenk flask equipped with a magnetic stirrer and sealed with a rubber septum. The mixture was subjected to three freeze−pump−thaw cycles to remove oxygen. Afterwards, the Schlenk flask under N_2_ was placed into a thermostatic bath at 140 °C for 30 min, followed by 48 h at 90 °C. Then, the reaction mixture was cooled down and diluted with a small amount of THF. The product was isolated by precipitation in a large amount of MeOH twice, filtered, and dried under vacuum for 24 h at 30 °C to give a yellow solid. Number average molecular weight (M_n_) determined by size exclusion chromatography (SEC) was 18,615 g/mol.

### 2.3. Synthesis of PS−b−PAN Copolymer

In a second step, the purified PS−CPDT macro-agent was used to synthesize PS−*b*−PAN copolymer. A typical polymerization procedure is described as follows: PS−CPDT macroCTA (0.1 g, 0.0056 mmol), AIBN (0.6 mg, 0.0038 mmol), and AN (2.9 mL, 44 mmol) were placed in a Schlenk flask equipped with a magnetic bar and dissolved in anhydrous DMF (6 mL). The tube was sealed with a rubber septum and the mixture was subjected to three freeze-pump-thaw cycles to remove oxygen. Then, the flask was placed into a thermostatic bath at 70 °C for 24 h under a N_2_ atmosphere. After this time, the reaction mixture was cooled down, precipitated in a large amount of MeOH, and filtered, repeated twice. Finally, the resulting block copolymer was dried under vacuum for 24 h at 30 °C as a white powder (M_n_ and polydispersity index (PDI) via SEC were 118,710 g/mol and 1.26, respectively).

### 2.4. Preparation of Porous Carbon Materials

Heat treatments were conducted in a horizontal tube furnace with a controlled atmosphere for both materials, block copolymer powder, and fibers. The as-obtained PS−*b*−PAN copolymer powder was heated up at a rate of 5 °C/min and isothermally stabilized at 280 °C for 1 h under air atmosphere. Then, the stabilized copolymer was carbonized at a rate of 5 °C/min at 800 °C for 1 h under N_2_ flow. The final powder carbon product was named as bulk material. Porous carbon fibers were produced from a solution of PS−*b*−PAN copolymer in DMF at 20 wt% concentration. The solution was stirred at 50 °C for 20 h and electrospun under the following parameters: flow rate of 1.5 mL/h, 15 cm of working distance, and a high voltage power supply of 18 kV under humidity ~40% RH at 20−25 °C. The fiber mat was collected on a stationary plate, peeled off from the aluminum foil, and dried in a vacuum oven at 50 °C for 5 h. The fibers were stabilized and carbonized by the same heat treatments as the block copolymer powders; air oxidation at 280 °C for 1 h and pyrolysis under N_2_ at 800 °C for 1 h.

### 2.5. Characterization

Molecular weights and blocks composition were measured by ^1^H−NMR in deuterated DMF with a Bruker DPX 300 MHz (Bruker, Rheinstetten, Germany) equipment. Molecular weight and polydispersity (PDI) of macroCTA and block copolymer were analyzed by Size Exclusion Chromatography (SEC) using a Waters 515 HPLC pump instrument (Waters, Barcelona, Spain) equipped with a Waters 24214 Refractive Index Detector and Agilent PLgel columns (500, 100, and mixed C). DMF at a flow rate of 0.5 mL/min was used as eluent at 45 °C working temperature and polystyrene (7,500,000−4490 g/mol) was utilized as standards for calibration (PolyScience (PolyScience, Niles, IL, USA)). Thermal transitions were analyzed by Differential Scanning Calorimetry (DSC) using a Mettler Toledo DSC SC822 equipment (Mettler Toledo, Madrid, Spain). Samples were placed in sealed aluminum pans purged with N_2_ flow using a heating and cooling rate of 10 °C min^−1^ from 25 to 350 °C. Glass transition temperature (*T_g_*) of the macroinitiator was determined from the second heating cycle. Thermogravimetric analysis (TGA) was carried out using TA instruments Q50 (TA instruments, New Castle, DE, USA) equipment. Measurements were performed with 8 mg of sample under N_2_ flow in a temperature range of 25−900 °C at a rate of 10 °C/min. According to Brunauer-Emmett-Teller (BET) method, surface areas were determined by N_2_ adsorption/desorption isotherms measured at 77 K on a Micromeritics ASAP-2020 (Micromeritics, Norcross, GA, USA). Samples were degassed at 200 °C under N_2_ atmosphere for 24 h prior to analysis. Pore size distributions were studied using Nonlocal Density Functional (NLDFT) method. The morphology and microstructure of the obtained porous carbon materials were imaged by Field Emission Scanning Electron Microscope (FESEM FEI TENEO LoVac (FEI, Hillsboro, OR, USA)) and Transmission Electron Microscope (TEM, Philips Tecnai 20 FEG (FEI, Hillsboro, OR, USA)). TEM samples were prepared by depositing the carbon material dispersion in ethanol into holey carbon copper grids. Raman spectroscopy (LabRAM HR800 spectrometer (Horiba, Kyoto, Japan) was performed with a 514.5 nm Ar laser excitation. X-ray photoelectron spectroscopy (SPECS GmbH with UHV system and energy analyzer PHOIBOS 150 9MCD (SPECS GmbH, Berlin, Germany) was carried out employing a non-monochromatic Al Mg X-ray source operated at 200 W.

Electrochemical characterization was evaluated using Biologic VSP-300 potentiostat (Biologic, Seyssinet-Pariset, France) working with a three electrodes cell configuration. Cyclic voltammetry, galvanic charge/discharge cycles, and electrochemical impedance spectroscopy measurements were carried out with Ag/AgCl as reference electrode, Pt as counter electrode, and aqueous KOH solution (6 M) as electrolyte. The carbon powder working electrode was prepared by pressing into a clean nickel foam a mixture of active material, black carbon, and PTFE 60% dispersion in water with 80:10:10 proportion, respectively. Fibers were tested as a self−standing electrode without the use of binder.

Capacitance values of the three−electrode cell were calculated from GCD and CV curve respectively using the following equations:(1)$$C_{s}=\frac{I\cdot \Delta t}{m\cdot \Delta V}$$
(2)$$C_{s}=\frac{1}{m\cdot \nu \cdot \Delta V}\int_{V_{0}}^{V_{t}}\left|I\left(V\right)\cdot dV\right|$$
where, Δ*V* is the voltage window, *m* is the active material electrode mass, *I* is the current used in the measure, Δt is the time to take place in the discharge curve, and ν is the sweep rate. For the symmetrical cell measurements, electrode capacitance value was calculated according to the equation below:(3)$$C_{s}=\frac{4\cdot I\cdot \Delta t}{m\cdot \Delta V}$$

As for the three−electrode cell measurements, Δ*V* corresponded with the electrolyte voltage window, *m* is the sum of the two−electrode active material mass, *I* is the discharge current, and Δt is the discharge time.

Energy density (W h/Kg) and power density (W/Kg) were calculated according to the following equations:(4)$$E\;\left(\mathrm{Wh}/\mathrm{Kg}\right)=\left(\frac{1}{2}\;C_{s}\;V^{2}\right)/3.6$$
(5)$$P\left(W/\mathrm{Kg}\right)=\frac{E\;\cdot \;3600}{\Delta t}$$
where ∆*t* (s) is the discharge time and *V* (V) is the discharge voltage range.

## 3. Results and Discussion

### 3.1. Synthesis of PS−b−PAN Copolymer

The copolymer PS−*b*−PAN, used as carbon templating material for fiber and bulk materials, was successfully synthesized by two-step RAFT polymerization (Appendix A). Figure 1a shows the ^1^H−NMR spectra of polystyrene macroCTA and block copolymer measured in *d_7_*-DMF. Signals related to macroCTA and copolymer were fully identified in the spectra. Molecular weight and degree of polymerization (DP) were determined by comparison of the relative integration of the signals at 2.45 ppm assigned to the protons −C*H_2_*−CH(CN)− of PAN and those at the range 6.7−7.4 ppm ascribed to the phenyl protons of PS. Calculations revealed the following composition: 0.16 volume fraction of polystyrene as sacrificial block (M_n, NMR_ = 16,403 g/mol) and 0.84 of polyacrylonitrile (M_n, NMR_ = 101,230 g/mol) as a carbon precursor block. SEC chromatograms presented unimodal narrow peaks for both macroCTA PS−CPDT and the BCP PS−*b*−PAN with low polydispersity index, 1.05 and 1.26, respectively (Figure 1b), indicating a successful control of the radical polymerization. Additionally, a large total molecular weight (sum of both blocks) has been obtained, which is demonstrated to be beneficial for strong block segregation [14,15] and facilitates fiber production by electrospinning.

### 3.2. Thermal Characterization

Bulk and fiber polymer materials were stabilized at 280 °C and carbonized at 800 °C under controlled conditions, after precipitation of the block copolymer and electrospinning, respectively. First, the morphology of the PAN phase is chemically fixed under air atmosphere through oxidative crosslinking reactions. In this step, side chain crosslinking and some cyclization occurred, converting C≡N into C=N bonds and turning them into thermally stables-triazine networks. Further cyclization followed by dehydrogenation (300–400 °C) and denitrogenation (>600 °C) leads to partially graphitic structures under an inert atmosphere [17,36]; whereas the sacrificial PS phase is thermally released as monomer in a gas phase, generating pores into the PAN carbon matrix [37]. PAN stabilization/oxidation processes were revealed in the DSC trace of block copolymer as a sharp exotherm at 260 °C (Figure 1c). The DSC trace of the macroCTA (PS−CPDT, inset in the Figure 1c) showed a *T_g_* at around 100 °C, corresponding to the amorphous region of PS, which is slightly visible in the PS−*b*−PAN thermogram. According to DSC data, the stabilization process was fixed at the upper limit temperature of the exothermic peak (280 °C for 2 h) to avoid a low efficiency of cyclization reactions and ensure an effective stabilization before PS decomposition. No melting peak of the PAN block was detected, due to low heating rates (<30 °C/min) oxidation/cyclization reactions occurred before melting [38]. Endothermic peaks corresponding to the decomposition of PS did not appear up to 350 °C, therefore, stabilization of PAN and decomposition of PS are well separated, which is highly desirable to preserve the morphologies generated in the microphase separation. Weight loss of PS−*b*−PAN due to pyrolysis was evaluated by TGA (Figure 1d). The TGA profile of PS-CPDT displayed a single weight-loss stage at 400 °C, while PS−*b*−PAN profile showed three loss stages. The first stage (~220–290 °C) corresponded to partial dehydrogenation and crosslinking (conversion of −C≡N into s-triazine ring) [39]. The second loss stage (~310−460 °C) was attributed to the thermal decomposition of the sacrificial PS block. The third stage showed a slight weight loss between 460 and 870 °C corresponding to further fragmental process of the pre-stabilized PAN [12,40]. The PS−*b*−PAN showed a 5% of weight loss at 290 °C in a nitrogen atmosphere and a char yield of 50%. Carbonization temperature was fixed at 800 °C for both bulk and fibers. Higher pyrolysis temperatures (>900 °C) are related to the introduction of quaternary nitrogen into the basal plane of graphitic structures, which do not significantly participate in electrochemical reactions [17].

### 3.3. Structural Characterization

The generated morphologies and microdomain sizes are affected by the phase separation between PS and PAN blocks [41]. Pore size is directly correlated to the domain size of the sacrificial block, which is related to molecular weight or degree of polymerization, Flory−Hugging’s interaction parameter (χ), and volume fraction (*f*) [42,43]. The fraction of sacrificial block is especially significant for designing block copolymers to obtain suitable porous carbon materials for capacitive energy storage [8]. In this work, PS has been chosen as a thermally sacrificial block based on its high incompatibility with PAN (segmental interaction parameter, χ_ij_ = 0.83 of acrylonitrile and styrene monomers) [44]. Due to fast solvent vaporization during electrospinning, BCP does not self−assemble into any thermodynamically equilibrated morphologies and, independent of composition, PS−*b*−PAN forms disordered nanostructures assembled through kinetic pathways, preventing any classical block copolymer thermodynamic equilibrated morphology such as spherical, cylindrical, or lamellar structures [45]. Instead, disordered, and interconnected domains are formed into the electrospun PS−*b*−PAN fibers. Similarly, due to the co−precipitation process with a non-solvent (MeOH) followed to obtain the powder material, disorder domains were also formed. Although the powder or bulk material can form conventional morphologies derived from self−assembly.

SEM and TEM images (Figure 2b,c, respectively) of the carbon bulk material exhibited a continuous surface porous morphology along the inner material, confirming the interconnectivity of the pores in the bulk grains. This type of disordered carbon structure with percolated pores was also formed with other PS−BCPs [21,46,47]. SEM images of the as−electrospun fibers from the PS−*b*−PAN copolymer showed rough surfaces (Figure 3b), typical of a fast self-assembling of the BCP during electrospinning. The average diameter of fibers corresponded to 107 ± 4 nm. After carbonization, fibers maintained their shape, although diameters hardly decreased (105 ± 5 nm). Fiber diameter distribution before and after carbonization can be found in Appendix A. SEM and TEM images (Figure 3c−e) revealed abundant mesopores on the surface and cross−section of the fiber, with an average pore size of around 10 nm (Figure 3f). Additional SEM and TEM images of carbon bulk and carbon fibers can be seen in Appendix A.

N−functionalities play an essential role in ensuring a rapid ion access into the pore structure. Pyridinic nitrogen, due to the combination of the lone pair and the π−electron system, leads to an enhancement of the ion conductivity and diffusion [48]. Thus, nitrogen exposed in the pore wall surfaces improves ion accessibility and, therefore, storage capability. In PAN−based BCPs, the interface between non−bonded PAN and PS domains contained pyridinic species that formed nitrogen−rich zigzag graphene edges (see schematic, Figure 2a) [17]. In disordered morphologies, such as those obtained here, non−bonded interfaces are predominant, guaranteeing a greater exposure of N during the electrochemical performance.

Raman spectroscopy provides information about ordered and disordered carbon structures. In Figure 3g, Raman spectra of mesoporous carbon bulk materials and fibers revealed the characteristic bands for highly ordered graphitic structures (“G band”) at ~1560−1600 cm^−1^ and disordered domains (“D band”) at ~1320−1350 cm^−1^ [49]. The intensity ratio of the D and G bands (*I_D_/I_G_*) may help elucidate the extent of carbon-containing defects and, therefore, to obtain information about the degree of graphitization of the structures. The *I_D_/I_G_* ratio reached a value of 1.11 for the fibers and 1.10 for the bulk powders, indicating a similar formation of graphitic structure for both materials. Deconvolution of the Raman spectra can be found in Appendix A. In addition, X-ray photoelectron spectroscopy (XPS) revealed the surface chemical composition of the carbon fibers. XPS full spectrum (Appendix A) confirmed the presence of three peaks corresponding to O 1s, N 1s, and C 1s at ~533.3, 401.3, and 284.6 eV, respectively. Sulfur peaks were not observed in the spectrum (S2s and S2p at ~240 and ~163 eV, respectively), which confirmed the elimination of sulfur−containing end−groups in the copolymer by pyrolysis. The carbon (Appendix A), oxygen, and nitrogen 1s XPS spectra can be resolved by curve fitting into several peaks with attributable binding energies [50,51,52,53] to estimate the presence of different types of functional groups. The N 1s spectrum was deconvoluted into four peaks (Figure 4a), verifying the existence of pyridinic nitrogen (N−P) at 398.2 eV, amide nitrogen (O=C−N) at 399.5 eV, pyrrolic/pyridone nitrogen (N−X) at 400.8 eV, and pyridine−N-oxide (N−O) at 402.7 eV. For the oxygen 1s deconvoluted spectrum (Figure 4b), the binding energies at 530.6, 532.5, 534, and 535.5 eV corresponded to quinone type groups (C=O; O−I), phenol and/or ether groups (C−OH and/or C−O−C; O−II), ester groups (O−C=O; O−III) and amide groups (N−C=O; O−IV), respectively, demonstrating a highly oxygen-rich material. Nitrogen atoms were intrinsically present in PAN, whereas oxygen atoms were originated during crosslinking under air atmosphere. Appendix A presented the relative surface concentrations of nitrogen and oxygen species determined by fitting the N 1s and O 1s core level spectra. Furthermore, the atomic percentage of N, C, and O was determined from XPS. As discussed above, the presence of heteroatoms may lead to a potential enhancement of charge mobility and electrolyte wettability. The latter was confirmed by measuring the contact angles of the fiber mats, before and after carbonization, with KOH aqueous solution (6 M). Images in Appendix A revealed that contact angle after pyrolysis was not noticeable due to the enhanced wettability of the material in presence of oxygen, which will improve the effective contact area between the electrode and the electrolyte [54].

In order to confirm and further study the overall porous structure showed in SEM and TEM, N_2_ adsorption/desorption analysis was conducted to test the specific surface area (SSA) and pore size distribution. Fiber material revealed a type IV isotherm with a large hysteresis loop (Figure 4c), which indicated the presence of mesopores, according to published IUPAC report [55]. Results were in good agreement with the pore structure previously observed in TEM images. Fast adsorption at the low relative pressure range (P/P_0_ < 0.1) confirmed the presence of micropores, revealing a BET SSA of 491.6 m^2^/g in which 170.4 m^2^/g corresponded to micropore SSA. In contrast, the bulk materials reached a BET specific surface area of 242 m^2^/g with micropore SSA of 93.9 m^2^/g and an isotherm type II (Figure 4c), suggesting a low pore density in the material. For both materials, pore sizes are concentrated in the mesopore and micropore range (Figure 4d). In the case of the fibers, pore size distribution with predominant micropore sizes and mesopores between 5 and 18 nm is shown. The presence of different micropore/mesopore sizes suggests a hierarchical pore structure within the material, which has been reported to be beneficial, offering transport pathways for ions, reducing diffusion distances from the electrolyte to the micropores, and providing a large number of active sites which results in a high-rate capacity [26]. According to the isotherm data, the bulk material exhibited lower pore volume and reduced the number of peaks in the mesopore range, indicating that block copolymer−derived porous carbon fiber showed a considerable better control over the pore formation and double value of specific surface area with respect to the bulk material.

### 3.4. Electrochemical Characterization

In order to determine the influence of the pore structure on their electrochemical performance, fibers and bulk material were tested in a three−electrode electrochemical cell. Cyclic voltammetry (CV) was evaluated at different sweep rates between 5 and 500 mV/s (Appendix A). Figure 5a showed the comparison of the electrochemical behavior of both materials (voltammetry at 5 mV/s), showing a rectangular shape close to an electric double layer capacitance (EDLC) but slightly distorted from an ideal supercapacitor, which evidenced how charge is not stored through a purely capacitive mechanism. This deviation from the ideal behavior can be influenced by the participation of pseudocapacitive storage mechanism involving fast and reversible redox reactions due to the edge N functionalities inherited from PAN stabilization process [56]. Consistent with the CV data, galvanostatic charge−discharge (GCD) curves of the fiber material displayed a triangular shape (Figure 5b) and a gravimetric capacitance value of 254 F/g measured at a current density of 1 A/g and 308 F/g measured at 10 mV/s. By contrast, bulk material exhibited a considerable reduced capacitance value of 145 F/g calculated at the same current density (1 A/g). The superior SSA_BET_, pore volume, and hierarchical porosity rich in micropores/mesopores presented in the fiber material resulted in an improvement in electrochemical storage. Electrochemical performance between various PCFs and carbon powder materials obtained through different templates using blends or copolymer as precursors is shown in Table 1.

Electrochemical impedance spectroscopy (EIS) was conducted from 0.1 to 100 KHz with a perturbance of 5 mV. Nyquist plots have been fitted with the equivalence circuit shown in Appendix A. For both materials, Nyquist diagrams (Figure 5c) can be divided into two different regions [57], one at high frequencies displaying a semicircle shape, from which values of R_S_ (series resistance) and R_CT_ (charge−transfer resistance) can be obtained, and a low frequency region showing a line shape related to the Warburg impedance. R_S_ is mainly associated with the intrinsic resistances of the electrode, electrolyte, and current collector. Both materials exhibited R_S_ values less than 1 Ω, with a value of 0.5 Ω for the fiber and 0.6 Ω for the bulk material. Note that the lower value obtained for the PCFs is due to both, its continuous structure with minimum interface effects, providing intimate contact with the electrolyte, and the intrinsic channels (or large pores) between individual fibers, which allows the electrolyte to reach the electrode core faster compared to the powder carbon.

**Table 1 polymers-14-05109-t001:** Summary of specific surface area and specific capacitance values measured at different current densities (A/g) of various porous carbon fibers (PCFs) and porous carbon powder materials as electrodes for supercapacitors.

Material	Precursor	SSA (m^2^/g)	Cs (F/g)	Electrolyte	Reference
**Powder carbons**					
S/N-doped porous carbons	PBA−*b*−PAN	478	236 (0.1 A/g)	KOH 6M	[58]
Open-ended hollow carbon spheres (HCS)	PS−*b*−P4VP/ KOH activation	1583	249 (0.5 A/g)	KOH 6M	[59]
N-doped hierarchical porous carbons (NHPC)	PS−*b*−PAN/KOH activation	2105	230 (1 A/g)	KOH 6M	[47]
Mesoporous carbons	PAN−*b*−PS−*b*−PAN	954	185 (0.625 A/g)	KOH 2M	[21]
**Fibers**					
Linear-tube carbon nanofibers (LTCNF)	PAN/PS	212	188 (0.5 A/g)	LiOH 3M	[60]
Multichannel PCFs	PAN/PS	750	250 (1 A/g)	KOH 6M	[61]
N-doped multi-nano-channel PCFs	PAN/PS	840	461 (0.25 A/g)	H_2_SO_4_ 1M	[62]
Mesoporous carbon nanofibers (MCNFs)	PAN/PAA−*b*−PAN−*b*−PAA	250	256 (0.5 A/g)	KOH 4M	[23]
PCFs	PAN/PMMMA	683	140 (0.5 A/g)	KOH 6M	[63]
Interconnected meso-PCFs	PMMA−*b*−PAN	503	360 (1 A/g)	KOH 6M	[34]
Hierarchical PCFs	PS−*b*−PAN	492	254 (1 A/g)	KOH 6M	This work

Additionally, R_CT_ mainly depends on the electronic and ionic resistances at the interfaces and in the whole system. Values of R_CT_ in fiber and bulk material corresponded with 0.45 and 0.60 Ω, respectively. The lower value achieved by the fibers is due to the synergistic effect of higher pore density and continuous structure, which improved the ion diffusion. Fiber presented a 50% capacitance retention at high sweep rates (500 mV/s), which is significantly higher than the 30% retention of the bulk material (Figure 5d).

To further assess more precisely the behavior of the fiber material, electrodes were tested in a symmetrical Swagelok type cell. CV curves (Figure 6a) presented a rectangular shape even at high sweep rates. Charge−discharge curves (Figure 6b) also exhibited a triangular shape showing a specific capacitance value of 234 F/g at a current density of 1 A/g, slightly lower than the capacitance value obtained in the three−electrode cell. EIS diagram (inset in Figure 6c) showed a considerable decrease in the Rs with a value of 0.25 Ω; however, R_CT_ value was the same as in the three−electrode cell measurements (0.5 Ω). Electrode stability was further evaluated. Capacitance remained stable without noticeable degradation along 10,000 cycles, demonstrating remarkable capacitance retention of 99.9% and electrode stability. Ragone plot of the fibers (Figure 6d) displayed a high energy density and power density of 20 W h/Kg and 12,300 W/Kg, respectively. 

## 4. Conclusions

Polystyrene and polyacrylonitrile copolymer (PS−*b*−PAN) of high molecular weight and low polydispersity has been successfully synthesized by RAFT polymerization. This copolymer had been proven to be a suitable precursor for the obtention of fibers and powder carbon materials. Due to the high incompatibility between PAN and PS blocks, a lower content of the sacrificial block is required (*f*_PS_ ≈ 0.16) for obtaining highly porous structures with a considerably high carbon yield (~50%). Fibers of PS−*b*−PAN produced by electrospinning combined with a pyrolysis process showed hierarchical porosity with mainly micro and mesopores (2−20 nm) and relatively high SSA (~492 m^2^/g), demonstrating a better control over pore formation than materials in bulk. The electrochemical behavior of both carbon materials was tested. As expected, fibers showed superior performance, almost doubling the capacitance value of the bulk material, 254 F/g and 145 F/g (measured at 1 A/g), respectively. Fibers also were tested as a free-standing electrode in a symmetrical cell showing excellent cycle stability. In short, this work provides valuable guidance for understanding the phase-separation behavior and the designing of block copolymers for their use as a template. Concretely, focusing on fibers−based carbon electrodes for energy storage devices and for other applications where high carbon-yield materials with good control over the porous structure are required.

## Data Availability

Not applicable.

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
