# Peer review of "Polyacrylonitrile-b-Polystyrene Block Copolymer-Derived Hierarchical Porous Carbon Materials for Supercapacitor"

_polymers, 2022, doi:10.3390/polym14235109_

Round 1

Reviewer 1 Report

The conversion of the block copolymer to porous carbon materials for supercapacitors is an interesting topic. The manuscript presents PS-b-PAN as a soft template for this purpose. However, the authors need to address the following comments;

The concept of converting block copolymer to porous carbon for supercapacitors is known. See, for example, Xinxing Tan Cailiao/New Carbon Materials

Volume 30, Issue 4, Pages 302 - 3091 August 2015; 

Journal of Materials Science

Volume 53, Issue 17, Pages 12115 - 121221 September 2018

Also, PS-B-PAN has been made before. See for example 

European Polymer Journal

Volume 119, Pages 102 - 113, October 2019

My main concern is, what is new in the present work?

Please cite references for the reported PS-b-PAN.

Also, please write a table of merit to compare the data obtained for porous carbon and its potential as a supercapacitor compared with other reported methods.

Author Response

The conversion of the block copolymer to porous carbon materials for supercapacitors is an interesting topic. The manuscript presents PS-b-PAN as a soft template for this purpose. However, the authors need to address the following comments:

The concept of converting block copolymer to porous carbon for supercapacitors is known. See, for example, Xinxing Tan Cailiao/New Carbon Materials

Volume 30, Issue 4, Pages 302 - 3091 August 2015.

Journal of Materials Science

Volume 53, Issue 17, Pages 12115 - 121221 September 2018

Also, PS-B-PAN has been made before. See for example

European Polymer Journal

Volume 119, Pages 102 - 113, October 2019

My main concern is, what is new in the present work?

Please cite references for the reported PS-b-PAN.

Answer: Thank you very much, we really appreciate your comments. We are aware that the use of block copolymers as templates for the obtention of porous carbon materials have been widely studied. We mention and reference it in the introduction (see lines 58-76). In particular, the use of PS-b-PAN as precursor has already been properly cited in the manuscript. For example, we have referenced the study of carbon porous materials (powder or bulk materials) obtained from de same block copolymer PS-b-PAN (Energy Storage Mater. 2016, 3, 140–148), numbered in the manuscript as reference 47; as well as the reference that you mentioned (New Carbon Materials 2015, 30, 302 – 3091), numbered as reference 21.

Thanks to the versatility that copolymers offer, we present the same copolymer but with a very different composition from those reported in the literature, and with the advantage of having a very low PS fraction (fPS ≈ 0.16), which allows to obtain materials with a high carbon yield. In addition, we have designed a new synthetic route based on RAFT polymerization to reach high molecular weights. Our main contribution in this work has been to fabricate fibers using PS-b-PAN as precursor. The use of block copolymers (BCPs) as precursors for the obtention of porous carbon fibers is a relatively new concept with great potential for energy store devices. Up until now, professor G. Liu’s group have reported the first and only study using as block copolymer PMMA-b-PAN (see reference 34 of the manuscript) to fabricate porous carbon fibers. In the manuscript, we mention it, and we compare our results with theirs. We consider that our work with another block copolymer contributes to enhance the knowledge in this field.

Also, please write a table of merit to compare the data obtained for porous carbon and its potential as a supercapacitor compared with other reported methods.

Answer: Thank you for your suggestion, we completely agree. We have included a table (Table 1) in the manuscript (see lines 410 and 413), comparing our work with different carbon porous fibers and powder materials for their use as electrodes in supercapacitors.

Reviewer 2 Report

The paper is interesting and the experimental data is well supported the conclusions. The paper is well written. I think this paper can be accepted in current form.

Author Response

The paper is interesting, and the experimental data is well supported the conclusions. The paper is well written. I think this paper can be accepted in current form.

Answer: We are delighted to note that you have considered the relevance of our work and we appreciate your generous consideration about it.

English language and style are fine/minor spell check required.

Answer: Thank you for carefully reading our manuscript. We have double checked the language and spelling. Some revisions are highlighted in yellow in the revised manuscript.

Reviewer 3 Report

The manuscript “Polyacrylonitrile-b-polystyrene block copolymer-derived hierarchical porous carbon materials for supercapacitor” reports the production of porous carbons as electrode materials in energy storage devices from block copolymer as a sacrificial template. The work is well organized and the data are supported by adequate statements. Some minor revisions are required:

1.      Which are the temperature and relative humidity of electrospinning process?

2.      The measure units of power density in eq. 5 are different from those reported in the text which are supposed to be wrong.

3.      Figure S2: the image is quite blurry, The x-axis should start from zero. The data seem to show a bimodal curve. Did the authors consider just the highest peak?

4.      Are the reported results of electrochemical performances comparable with similar systems reported into the literature? A comparison with other works should be briefly reported in the last section.

Author Response

The manuscript “Polyacrylonitrile-b-polystyrene block copolymer-derived hierarchical porous carbon materials for supercapacitor” reports the production of porous carbons as electrode materials in energy storage devices from block copolymer as a sacrificial template. The work is well organized, and the data are supported by adequate statements. Some minor revisions are required:

  1. Which are the temperature and relative humidity of electrospinning process?

Answer: Thank you very much, we really appreciate your comment. Both temperature and humidity were used in the laboratory conditions, and they were not controlled. Even though, they were measured during the experiments with values for temperature and humidity ranging from 20 to 25 ºC and closed to 40% RH, respectively. To clarify this, we have included that information in the manuscript, see lines 144-146.

  1. The measure units of power density in eq. 5 are different from those reported in the text which are supposed to be wrong.

Answer: Thank you very much for your exhaustive revision. As you mentioned, the units in equation number 5 were wrong. We have corrected that mistake, see line 200.

  1. Figure S2: the image is quite blurry, the x-axis should start from zero. The data seem to show a bimodal curve. Did the authors consider just the highest peak?

Answer: Thank you for the recommendation regarding the statistical treatment of the fiber diameter data. We have improved the graph distribution and the fitting function, which fits better to the experimental data without requiring a bimodal distribution. So, average fiber diameter was determined by the maximum of the fitting distribution, showed in figure S2. The data has been corrected in lines 286-288 of the manuscripts as well.

  1. Are the reported results of electrochemical performances comparable with similar systems reported into the literature? A comparison with other works should be briefly reported in the last section.

Answer: Thank you for your suggestion, we completely agree. We have included a table (Table 1) in the manuscript (see lines 410 and 413), comparing our work with different carbon porous fibers and powder materials for their use as electrodes in supercapacitors.

Moderate English changes required.

Answer: Thank you for carefully reading our manuscript. We have double checked the language and spelling. Some revisions are highlighted in yellow in the revised manuscript.

Round 2

Reviewer 1 Report

The authors have made the proper revision.